# Risks of Stroke and Transient Cerebral Ischemia up to 4 Years Post-SARS-CoV-2 Infection in Large Diverse Urban Population in the Bronx

**DOI:** 10.3390/diagnostics15243183

**Published:** 2025-12-13

**Authors:** Sagar Changela, Roham Hadidchi, Aditi Vichare, Liora Rahmani, Sonya Henry, Tim Q. Duong

**Affiliations:** Department of Radiology, Albert Einstein College of Medicine and Montefiore Health System, Bronx, NY 10461, USA; sagar.changela@einsteinmed.edu (S.C.); roham.hadidchi@einsteinmed.edu (R.H.); avichare@g.ucla.edu (A.V.); liorarahmani1@gmail.com (L.R.); sonya.henry@einsteinmed.edu (S.H.)

**Keywords:** SARS-CoV-2, ischemic stroke, transient cerebral ischemia, COVID-19 outcomes, cerebrovascular risks

## Abstract

**Background:** SARS-CoV-2 infection could trigger hypercoagulation and hyperinflammation that may predispose patients to cerebrovascular events. The long-term risk of stroke among COVID-19 patients remains unclear. This study investigated the long-term risks of ischemic stroke and transient cerebral ischemia (TCI) among patients with and without COVID-19. **Methods:** We conducted an observational cohort study in the Montefiore Health System (February 2020–January 2024), with 52,117 COVID+ and 837,395 COVID− patients without prior cerebrovascular events. Demographics, comorbidities, insurance, unmet social needs, and median income were adjusted for using inverse probability weighting. Cox-proportional regression hazard ratios (HR) and their 95% confidence intervals were computed for ischemic stroke and TCI. **Results:** Compared to COVID− controls, ischemic stroke risk was higher among hospitalized COVID+ patients (HR = 1.32 [1.12–1.55]) and non-hospitalized COVID+ patients (1.21 [1.05–1.39]). Compared to COVID− controls, TCI risk was similar among hospitalized COVID+ patients (1.00 [0.75–1.33]), but higher among non-hospitalized COVID+ patients (2.15 [1.81–2.56]). **Conclusions:** Hospitalized and non-hospitalized COVID-19 patients had a higher long-term risk of ischemic stroke while only non-hospitalized COVID-19 patients had a higher long-term risk of TCI. These findings underscore the needs for long-term monitoring of cerebrovascular risk factors in COVID-19 survivors.

## 1. Introduction

SARS-CoV-2 is associated with a wide spectrum of acute and subacute neurological syndromes, broader than those seen with other viral infections [1,2,3]. Documented severe neurologic complications of acute COVID-19 include ischemic stroke, intracranial hemorrhage, transient cerebral ischemic encephalitis, and neuromuscular disorders [4,5]. In addition, many patients experience persistent neurocognitive symptoms [6,7], including memory loss, brain fog, attention deficits, and executive dysfunction, long after the resolution of acute infection. During acute COVID-19, direct viral invasion of endothelial cells, cytokine-mediated inflammation, and disruption of the renin–angiotensin–aldosterone system can compromise vascular integrity and cerebral perfusion. These changes occur alongside marked coagulopathy, elevated D-dimer, endothelial activation, and microvascular thrombosis, all of which create favorable conditions for ischemic stroke or transient cerebral ischemia [8].

Emerging evidence suggests that COVID-19 may also predispose individuals to delayed cerebrovascular events, including ischemic stroke, transient cerebral ischemia (TCI), and intracranial hemorrhages, even months after clinical recovery [9,10]. These delayed events may result from lingering vascular injury, immune dysregulation, and a prolonged hypercoagulable state. Endothelial dysfunction, chronic inflammation, and elevated levels of pro-thrombotic markers such as fibrinogen and cytokines can persist beyond the acute phase, quietly compromising cerebrovascular health [11,12,13,14,15,16,17]. The occurrence of overt neurological injury in some patients raises concern that more subtle or progressive cerebrovascular damage may be occurring in others, even in the absence of obvious symptoms. Some studies have reported increased stroke and TCI risk within a year post-infection [18,19,20,21,22]. Data beyond this time window and the effects of other confounders (such as COVID-19 disease severity, COVID-19 vaccination and social determinants of health) remain sparse.

To address these knowledge gaps, we conducted a large observational cohort study in a racially and socioeconomically diverse urban population in the Bronx. We evaluated the long-term risks of first-time ischemic stroke and TCI up to four years after SARS-CoV-2 infection, using inverse probability weighting (IPW) to adjust for demographics, comorbidities, vaccination status, and social determinants of health. We stratified patients by COVID-19 hospitalization status to evaluate differential risks based on disease severity. Our findings provide new insight into the cerebrovascular sequelae of COVID-19 and highlight vulnerable populations who may benefit from long-term neurological surveillance.

## 2. Materials and Methods

### 2.1. Data Sources

This retrospective study was approved by the Einstein-Montefiore Institutional Review Board with waived informed consent (#2021-13658). Data came from the electronic health records (EHR) of the Montefiore Health System, which consists of multiple hospitals and outpatient clinics in the Bronx and its environs. Patents came to our healthy system for any medical reasons, including regular check-ups and screening for COVID-19. Data were extracted as previously described [23,24,25,26,27,28,29].

### 2.2. Study Cohort

The EHR was queried for all polymerase chain reaction (PCR) SARS-CoV-2 tests performed in the Montefiore Health System from 1 February 2020, to 12 January 2024. COVID+ patients consisted of those who tested positive at least once and index date was defined as date of first positive test. COVID− patients consisted of those who tested negative and index date was defined as first visit since 1 March 2020. Patients with a history of cerebrovascular accidents (ischemic stroke, TCI, or intracranial hemorrhage) and those lost to follow-up within 14 days of the index date were excluded (i.e., it excluded patients who died of cerebrovascular events in the acute phase (days 0–14 and, thus, this study measured the risk of stroke and TCI from COVID-19 in survivors beyond 14 days).

### 2.3. Variables

Demographic data included age at index date, sex, race, and ethnicity. Insurance and median household income of each patient’s Zone Improvement Plan (ZIP) code.

Pre-existing comorbidities at index date included type-2 diabetes (T2DM), hypertension (HTN), chronic obstructive pulmonary disease (COPD), chronic kidney disease (CKD), cardiovascular disease (CVD; defined as a composite of a history of myocardial infarction, coronary artery disease, or congestive heart failure), and asthma. COVID+ patients were also stratified by hospitalization due to acute COVID-19.

### 2.4. Outcomes

Outcome events included ≥ 1 diagnostic codes for ischemic stroke and TCI recorded in the EHR after the index date. Concept names and IDs for outcomes are listed in Appendix A. For all outcomes, follow-up time was calculated in months from the index date to either the date of first diagnosis (for patients who developed the outcome) or to the date of death or last recorded visit (for patients who did not develop the outcome) up to 12 January 2024. New hemorrhagic stroke was not evaluated due to very low incidence.

### 2.5. Statistical Analysis

IPW was used to balance the covariates between the groups. We selected inverse probability weighting because it provides efficient covariate balance across the three study groups: hospitalized COVID-19+, non-hospitalized COVID-19+, and COVID-19– controls, without reducing sample size. A multinomial logistic regression model was used to estimate the probability of a participant belonging to the observed group (COVID+ hospitalized, COVID+ non-hospitalized and COVID−) conditional on all pre-defined covariates listed in the variable section. The estimated probability of being in the observed group given covariates (P(group = observed group|L)) was used as the propensity score to derive inverse probability weights for estimating the average treatment effect within the cohort. Stabilized weights were calculated as the ratio of the overall group proportion (P(group = observed group)) to the propensity score, with the group proportion serving as the stabilization factor. After applying these weights, covariate balance across groups was evaluated using standardized mean differences [30]. Risks of outcomes were assessed using Cox-proportional hazards models and Kaplan–Meier survival curve formed to visualize the results. Univariate models were run on the IPW-adjusted data. All covariates were properly balanced across groups after weighing. Python (version 3.8.19) and R (version 4.4.0) were used for data processing and statistical analysis. *p*-values less than 0.05 were considered statistically significant. We also performed a subgroup analysis stratified by age, sex, race, ethnicity, ZIP code median income quartile, insurance coverage, comorbidity presence, and COVID-19 vaccination status.

## 3. Results

Figure 1 shows the patient selection flowchart. From 1 February 2020 to 12 January 2024, 1,299,852 patients visited the Montefiore Health System. After excluding patients with past medical history of cerebrovascular events, 59,179 COVID+ and 994,721 COVID− patients were identified. There were 52,117 COVID+ and 837,395 COVID− patients who returned to the Montefiore Health System 14 or more days after index date.

Table 1 shows the baseline characteristics of the three study groups: hospitalized COVID-19+, non-hospitalized COVID-19+, and COVID-19– patients. COVID+ hospitalized patients were on average older (61.64 vs. 39.60 vs. 42.01 years old), were more likely to be male (46.42% vs. 39.25% vs. 43.42%) and be on Medicare (33.44% vs. 7.49% vs. 11.57%) and all major pre-existing comorbidities compared to COVID+ non-hospitalized and COVID− patients. COVID+ non-hospitalized patients were less likely to be on Medicaid, less likely to be Hispanic, more likely to live in lower-income ZIP codes, and more likely to have all major pre-existing comorbidities compared to COVID− patients. Those hospitalized for COVID-19 had higher unadjusted incidence of ischemic stroke (2.87%) compared with non-hospitalized COVID-19 (0.85%) and COVID− controls (1.02%). Similarly, incidence of TCI was highest among hospitalized COVID-19 patients (0.77%), compared with non-hospitalized COVID-19 (0.50%) and COVID− (0.38%).

Table 2 shows the baseline characteristics of the pseudopopulation after IPW. Groups were balanced across all key covariates with absolute standardized differences of 0.1 or less in most variables.

Table 3 illustrates the results of Cox-proportional hazards models on ischemic stroke and TCI. For the entire period, COVID+ hospitalized patients (HR 1.32 [95% CI 1.12–1.55]) band COVID+ non-hospitalized patients (HR 1.15 [(95% CI 1.01–131]) were at the higher risk of ischemic stroke as compared to COVID- patients. Similarly, the risk of transient cerebral ischemia is higher among COVID+ non-hospitalized as compared to COVID− patients (HR 2.05 [95% CI 1.74–2.42]). However, the risk of transient cerebral ischemia is similar between COVID+ hospitalized and COVID− patients (HR 1.00 [95% CI 0.75–1.33]). The hazard ratios for different sub-periods are also shown. Figure 2 represents the Kaplan -Meier curves for outcome ischemic stroke and TCI.

A subgroup analysis stratified by age, sex, race, ethnicity, ZIP code median income quartile, insurance coverage, comorbidity presence, and COVID-19 vaccination status is shown in Appendix A. Across most subgroups, non-hospitalized COVID-19+ patients showed an increased risk of TCI compared with controls, and increased stroke risk was observed among those younger than 60, female, Hispanic, insured through Medicare, or with hypertension, asthma, or chronic kidney disease. In contrast, hospitalized COVID-19+ patients showed higher stroke risk only in select subgroups, including individuals older than 60, males and females, Black and Hispanic patients, those covered by Medicaid, and those with hypertension or asthma. Nearly all subgroups demonstrated no association between hospitalized COVID-19 positivity and TCI. The most notable finding was that among COVID-19 unvaccinated patients, non-hospitalized COVID-19 was associated with increased risk of stroke and TCI. But this association was not found among vaccinated patients.

## 4. Discussion

In this retrospective cohort study, we found that both hospitalized and non-hospitalized COVID+ patients exhibited significantly increased adjusted risk of first-time ischemic stroke compared to COVID− controls. However, only non-hospitalized COVID+ patients demonstrated a significantly elevated risk of TCI.

A recent UK Biobank study with up to three years of follow-up reported increased risk of major adverse cardiovascular events, including stroke, after COVID-19 (HR = 2.09 [1.94–2.25] overall; HR = 3.85 [3.51–4.24] hospitalized), reinforcing that SARS-CoV-2 infection confers a sustained thrombotic burden [31]. There are a few similar studies that have tracked ischemic stroke risk within a year post-COVID-19. These studies have reported a higher incidence of ischemic stroke among COVID-19 survivors compared with non-COVID-19 controls, with a meta-analysis indicating a two-fold increase in stroke risk (HR = 2.06 [1.76, 2.42]) persisting for up to a year post-infection [32]. A large observational study using United States Department of Veterans Affairs data (*n* = 154,068 COVID+ patients; 12-month follow-up) reported an increased risk of ischemic stroke among both hospitalized (HR = 3.18 [2.76, 3.66]) and non-hospitalized (HR = 1.27 [1.19, 1.36]) COVID+ patients [22,33]. Another retrospective cohort study using data from the United States Collaborative Network on TriNetX (*n* = 691,455 COVID+ patients; up to 12-month follow-up) found elevated risk for ischemic stroke (HR = 1.62 [1.55, 1.69]) among COVID-19 survivors compared to controls [19]. A prospective study of the U.K. Biobank (*n* = 17,871 COVID+ patients; 5-month follow-up) found that individuals hospitalized for COVID-19 had a significantly elevated risk of stroke (HR = 17.5 [5.26, 57.9]), whereas those not hospitalized did not exhibit increased risk [21]. The extraordinarily high hazard ratio might reflect the early pandemic, during which vaccines and effective treatment for COVID-19 were not yet available. Methodological differences, including outcome definitions, exclusion or inclusion of patients with prior cerebrovascular events, population demographics, follow-up time, and covariate adjustment, may have contributed to the heterogeneity of the findings across different studies to date. In particular, we performed vigorous controls using IPW and multivariable analysis for the perspective patients who were hospitalized for COVID-19 and those were not. Our study included up to four years of follow-up, in contrast to earlier studies with shorter observation windows.

We did not analyze outcomes with respect to different viral variant because SARS-CoV-2 strains were not tested on individual patients. The prevalence of variants was based on population level, and there were significant overlaps of the predominant variants at any given time, and thus defining predominant variants that patients were affected over a period is challenging, somewhat arbitrary and inaccurate. We thus did not analyze outcomes with respect to variants of the SARS-CoV-2 virus. Distinct variants of the SARS-CoV-2 virus are not available in the EHR. Nonetheless, the risk of ischemic stroke may vary across SARS-CoV-2 variants, reflecting differences in viral pathogenicity and the degree of associated inflammation and coagulopathy. Early variants such as Alpha and Delta were linked to higher rates of thrombotic complications, including stroke, likely due to more pronounced endothelial injury and hypercoagulability. In contrast, Omicron has generally shown lower neurological and thrombotic involvement, despite its higher transmissibility. These variant-specific differences underscore the evolving cerebrovascular impact of COVID-19 over the course of the pandemic [34,35,36]. However, it is important to note that the effects of variants to outcomes were likely confounded by available vaccines, available treatments, and other factors and are difficult to segregate.

The higher hazard ratio for TCI in non-hospitalized COVID-19 survivors, contrasted with the null association in the hospitalized group, appears counterintuitive. Several factors may explain this pattern. First, hospitalized patients with severe COVID-19 often receive early anti-inflammatory and antithrombotic therapies, which may mitigate subsequent vascular injury and reduce the likelihood of post-acute cerebrovascular events. Second, severe COVID-19 is associated with high short-term mortality; individuals at greatest risk for cerebrovascular complications may not have survived long enough to develop TCI, creating a competing-risk bias that attenuates associations in the hospitalized group. Third, diagnostic and coding patterns may differ between groups: hospitalized patients typically undergo comprehensive inpatient neurological evaluation and may have residual symptoms managed under existing diagnoses, whereas non-hospitalized patients who later present with transient focal deficits are more likely to receive a new TCI code. Finally, the number of TCI events was modest, and given multiple subgroup analyses, some of the observed heterogeneity may reflect chance. Together, these explanations provide plausible context for the observed differences while acknowledging the limitations inherent to EHR-based analyses. Moreover, TCI, in particular, may be more sensitive to subtle or transient vascular insults than ischemic strokes. While stroke typically results from more severe, sustained vascular injury, TCI may reflect temporary cerebrovascular dysfunction, potentially exacerbated by COVID-19-related inflammation or endothelial damage. In individuals with fewer competing risk factors, COVID-19-induced or related subclinical processes, such as persistent endothelial activation, low-grade inflammation, and delayed thrombotic sequelae, may be more likely to manifest as symptomatic TCI [15,37,38]. A prior study found increased TCI risk post-COVID (HR = 1.49 [1.37, 1.62]) [33], consistent with our main analysis. Another population-based study in Singapore however found no increase in TCI risk among non-hospitalized COVID-19 patients [39]. Notably, our extended four-year follow-up allowed for detection of delayed events that might have been missed in earlier studies with shorter observation windows.

Several biological mechanisms could explain the observed differences in long-term ischemic stroke and TCI risk. First, cardiopulmonary stress induced by SARS-CoV-2 infection can lead to hypoxia, placing significant strain on the cerebrovascular system. While this stress may not immediately result in acute stroke, it can manifest downstream as chronic endothelial dysfunction, impairing the neurovascular unit’s ability to maintain homeostasis. Hypoxia-driven stress may exacerbate pre-existing vulnerabilities, particularly in individuals with underlying cardiopulmonary conditions, and contribute to long-term cerebrovascular damage [14,40,41]. Second, SARS-CoV-2 infection is associated with a hypercoagulable state, characterized by elevated levels of fibrinogen, von Willebrand factor, and platelet activation [37,42,43]. This pro-thrombotic environment can promote the formation of microthrombi and larger thrombi, as well as worsen existing plaque buildup or clots. While these effects also may not always manifest acutely, they can lead to delayed cerebrovascular events, such as ischemic stroke or TCI, months to years after the initial infection [44,45,46]. Third, systemic hyperinflammation driven by elevated concentrations of cytokines such as interleukin-6 and tumor necrosis factor-α can disrupt vascular homeostasis, promote plaque instability, and trigger adverse vascular remodeling [47,48,49]. This inflammatory milieu overlaps with the hypercoagulable state, further exacerbating endothelial injury and increasing the risk of ischemic events. Elevated D-dimer and ferritin levels observed during acute COVID-19 may contribute to ongoing vascular injury and prothrombotic activity, providing a plausible mechanism for delayed ischemic stroke or TCI [50,51]. Additionally, large longitudinal studies have shown that dyslipidemia, chronic inflammation, and their cumulative burden substantially elevate long-term cardiovascular and cerebrovascular risk, supporting the concept that persistent vascular and immune dysregulation, such as that seen following SARS-CoV-2 infection can predispose individuals to ischemic events [52,53,54]. The interplay of cardiopulmonary stress, hypercoagulability, and hyperinflammation converges to induce endothelial injury within the neurovascular unit [47,48,55]. This injury can impair blood–brain barrier integrity, reduce vasodilation capacity, and foster a pro-thrombotic environment, ultimately increasing the risk of ischemic stroke and TCI [56,57]. The stronger and more consistent association observed for ischemic stroke across both hospitalized and non-hospitalized COVID+ patients suggests that SARS-CoV-2 may induce lasting vascular injury severe enough to provoke permanent ischemia even among individuals with differing baseline comorbidities. In contrast, the selective increase in TCI risk among non-hospitalized patients may reflect more subtle or reversible cerebrovascular dysfunction, which becomes detectable primarily in individuals with otherwise lower baseline vascular burden. Further research is needed to elucidate how these pathways interact across varying severities of COVID-19 and to identify targeted interventions that mitigate the cerebrovascular sequelae of SARS-CoV-2 infection.

This study has several limitations. EHR could have misclassification. We restricted inclusion to PCR-confirmed COVID-19 cases, as home test results were inconsistently documented, and some patients may have tested positive outside our system. Consequently, misclassification of COVID-19 status may have occurred, with undetected infections being included in the COVID-negative group. This would likely bias our effect estimates toward the null. Stroke and TCI were identified using diagnostic codes, which may introduce misclassification, especially for TCI, where diagnosis is more variable. This limitation could contribute to differences observed between the two outcomes, and future work with validated or clinically confirmed endpoints would help strengthen these findings. COVID-19 vaccination status was not analyzed with respect to outcomes, given that many patients likely received vaccines at external locations (e.g., local pharmacies). As vaccination reduces the risk of severe outcomes, its potential effect on post-COVID cerebrovascular risk warrants future investigation. We also did not analyze the influence of specific COVID-19 treatments on outcomes. Treatment regimens, including antivirals, steroids, and anticoagulants, evolved over time and were inconsistently prescribed especially during early pandemic. We were unable to adjust for certain stroke-specific risk factors, including blood pressure, atrial fibrillation, and antithrombotic or lipid-lowering medication use, which may contribute to residual confounding. Our cohort is diverse, and our findings may not apply to less diverse populations. Although we employed inverse probability weighting and Cox proportional hazard regression, residual confounding and selection bias remain inherent limitations of observational studies. While the Kaplan–Meier curves show clear separation between groups, substantial attrition in the COVID-positive cohorts at later time points increases uncertainty in survival estimates. Loss to follow-up may also be non-random, introducing the possibility of informative censoring. Therefore, results toward the tail of the curves should be interpreted with caution. Finally, although the extended four-year follow-up enabled detection of delayed cerebrovascular events, differences in stroke and TCI risk patterns between hospitalized and non-hospitalized cohorts suggest that unmeasured factors—such as differential health-seeking behaviors, subclinical comorbidity burden, or variation in SARS-CoV-2 strain exposure—may have contributed to the observed associations.

## 5. Conclusions

Both hospitalized and non-hospitalized COVID+ patients exhibited a higher risk of new-onset ischemic stroke compared to matched COVID− controls over up to four years of follow-up. Non-hospitalized COVID+ patients also showed an increased risk of transient cerebral ischemia, whereas hospitalized COVID+ patients showed similar risk compared to COVID− controls. These findings underscore the importance of long-term monitoring for cerebrovascular complications among high-risk COVID-19 survivors.

## Figures and Tables

**Figure 1 diagnostics-15-03183-f001:**
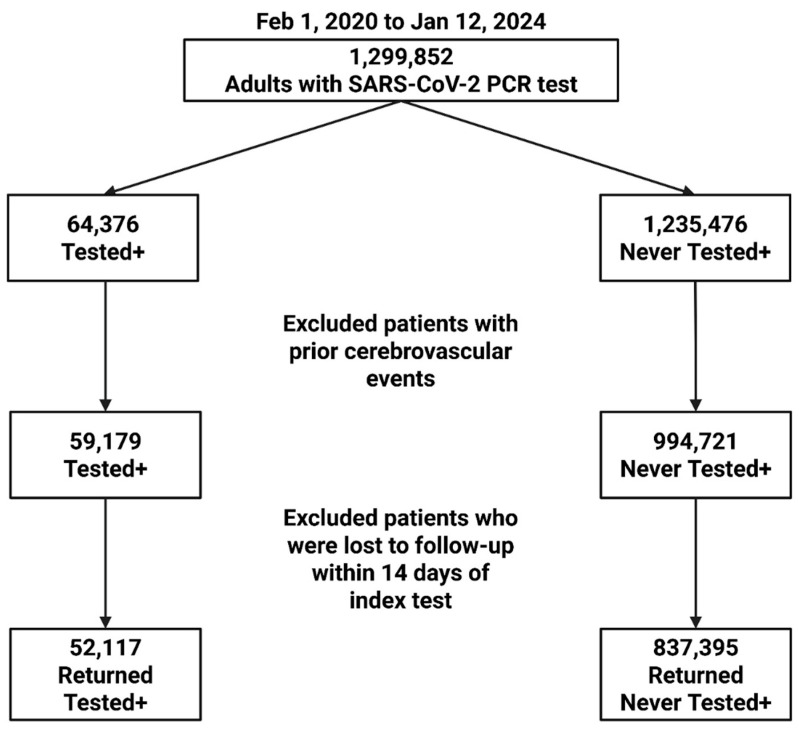
Patient selection flowchart.

**Figure 2 diagnostics-15-03183-f002:**
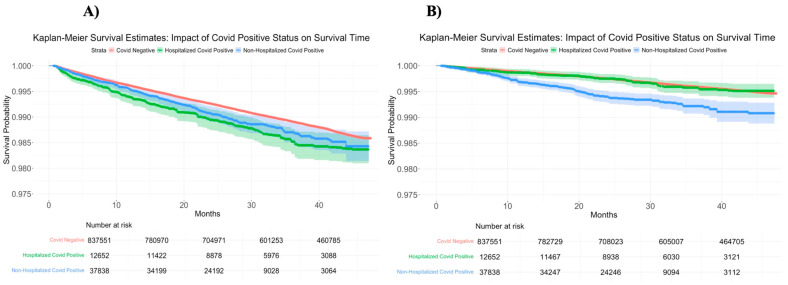
Kaplan–Meier survival curve for outcome (**A**) ischemic stroke and (**B**) transient cerebral ischemia.

**Table 1 diagnostics-15-03183-t001:** Baseline characteristics of patients prior to weighting, excluding those with a history of cerebrovascular events. Absolute standardized differences were calculated to assess covariate balance across the groups: A vs. C: Hospitalized COVID+ vs. COVID−; B vs. C: Non-Hospitalized COVID+ vs. COVID−; A vs. B: Hospitalized COVID+ vs. Non-Hospitalized COVID+.

Covariates	Hospitalized COVID+ *N* = 13,983	Non-Hospitalized COVID+ *N* = 38,134	COVID− *N* = 837,395	Absolute Standardized Difference		
				A vs. C	B vs. C	A vs. B
**Age, mean ± standard deviation**	61.64 ± 20.48	39.60 ± 22.89	42.01± 24.20	0.85	0.1	0.98
**Age** categorical ≥ 60 years	8448 (60.41%)	8391 (22.01%)	232,917 (27.81%)	0.69	0.14	0.85
**Male, n (%)**	6491 (46.42%)	14,970 (39.25%)	363,624 (43.42%)	0.06	0.08	0.15
**Race, n (%)**						
White	1858 (13.29%)	4028 (10.56%)	111,584 (13.32%)	0.001	0.08	0.08
Black	4764 (34.07%)	11,581 (30.37%)	223,519 (26.69%)	0.16	0.08	0.08
Asian	470 (3.36%)	1830 (4.80%)	25,125 (3.00%)	0.02	0.09	0.07
Others	6891 (49.28%)	20,695 (54.27%)	477,167 (57.00%)	0.15	0.05	0.1
**Ethnicity** Hispanic, n (%)	6767 (48.39%)	20,706 (54.30%)	482,816 (57.65%)	0.19	0.07	0.12
**ZIP Code Median Income, n (%)**						
1st Quartile (0–25)	3481 (24.89%)	10,504 (27.54%)	253,843 (30.31%)	0.12	0.06	0.06
2nd Quartile (25–50)	3992 (28.55%)	10,044 (26.34%)	197,267 (23.56%)	0.11	0.06	0.05
3rd Quartile (50–75)	4205 (30.07%)	10,169 (26.67%)	204,023 (24.37%)	0.13	0.05	0.07
4th Quartile (75–100)	2305 (16.48%)	7417 (19.45%)	182,262 (21.76%)	0.13	0.06	0.07
**Insurance, n (%)**						
Medicare	4676 (33.44%)	2858 (7.49%)	96,880 (11.57%)	0.54	0.14	0.68
Medicaid	5253 (37.57%)	17,556 (46.06%)	378,601 (45.21%)	0.16	0.02	0.17
Private	3621 (25.90%)	13,441 (35.25%)	281,785 (33.64%)	0.17	0.03	0.21
Self-pay	433 (3.10%)	4279 (11.22%)	80,129 (9.57%)	0.27	0.05	0.32
**Vaccination (at least one dose)**	5824 (41.70%)	20,245 (53.10%)	312,152 (37.30%)	0.09	0.32	0.23
**Comorbidities, n (%)**						
Type-2 Diabetes	7110 (50.85%)	8385 (21.99%)	118,590 (14.16%)	0.85	0.2	0.63
Hypertension	9340 (66.80%)	10,753 (28.20%)	170,682 (20.39%)	1.06	0.18	0.84
COPD	1786 (12.77%)	892 (2.34%)	12,482 (1.49%)	0.45	0.06	0.40
Chronic Kidney Disease	4214 (30.14%)	2282 (5.98%)	33,623 (4.02%)	0.74	0.09	0.66
Cardiovascular Disease	4747 (33.95%)	2751 (7.21%)	43,799 (5.23%)	0.77	0.08	0.70
Asthma	3150 (22.53%)	7630 (20.01%)	87,846 (10.49%)	0.33	0.27	0.06
**Outcome**						
Stroke	401 (2.87%)	326 (0.85%)	8546 (1.02%)	-	-	-
Transient Cerebral Ischemia	107 (0.77%)	192 (0.50%)	3175 (0.38%)	-	-	-

**Table 2 diagnostics-15-03183-t002:** Baseline characteristics of patients after inverse probability weighting, excluding those with a history of cerebrovascular events. Absolute standardized differences were calculated to assess covariate balance across the groups: A vs. C: Hospitalized COVID+ vs. COVID−; B vs. C: Non-Hospitalized COVID+ vs. COVID−; A vs. B: Hospitalized COVID+ vs. Non-Hospitalized COVID+.

Covariates	Hospitalized COVID+ *N* = 12,652	Non-Hospitalized COVID+ *N* = 37,838	COVID− *N* = 837,551	Absolute Standardized Difference		
				A vs. C	B vs. C	A vs. B
**Age, mean ± standard deviation**	44.04 (**±**23.68)	41.59 ((**±**23.16)	42.22 (**±**24.30)	0.07	0.03	0.1
**Age** categorical ≥ 60 years	3556 (28.1%)	9236 (24.4%)	236,740 (28.3%)	0.003	0.09	0.08
**Male, n (%)**	5094 (40.3%)	16,637 (43.9%)	362,844 (43.3%)	0.06	0.01	0.07
**Race, n (%)**						
White	1596 (12.6%)	4825 (12.7%)	110,592 (13.2%)	0.02	0.01	0.003
Black	3524 (27.9%)	10,404 (27.4%)	225,901 (27.0%)	0.02	0.01	0.009
Asian	382 (3.0%)	1174 (3.1%)	25,829 (3.1%)	0.004	0.001	0.004
Others	7148 (56.5%)	21,498 (56.7%)	475,231 (56.7%)	0.005	0.001	0.004
**Ethnicity** Hispanic, n (%)	7356 (58.1%)	21,647 (57.1%)	480,434 (57.4%)	0.02	0.005	0.02
**ZIP Code Median Income, n (%)**						
1st Quartile (0–25)	4093 (32.4%)	11,458 (30.2%)	252,189 (30.1%)	0.05	0.003	0.05
2nd Quartile (25–50)	3028 (23.9%)	9161 (24.2%)	198,986 (23.8%)	0.004	0.01	0.006
3rd Quartile (50–75)	3073 (24.3%)	9323 (24.6%)	205,656 (24.6%)	0.006	0.001	0.007
4th Quartile (75–100)	2456 (19.4%)	7960 (21.0%)	180,721 (21.6%)	0.05	0.01	0.04
**Insurance, n (%)**						
Medicare	1649 (13.0%)	4202 (11.1%)	98,418 (11.8%)	0.04	0.02	0.06
Medicaid	5677 (44.9%)	17,186 (45.3%)	377,910 (45.1%)	0.005	0.004	0.009
Private	4423 (35.0%)	12,953 (34.2%)	281,384 (33.6%)	0.03	0.01	0.02
Self-pay	901 (7.1%)	3561 (9.4%)	79,841 (9.5%)	0.09	0.005	0.08
**Vaccination (at least one dose)**	4907 (38.8%)	14,503 (38.3%)	318,470 (38.0%)	0.02	0.006	0.009
**Comorbidities, n (%)**						
Type-2 Diabetes	2543 (20.1%)	5797 (15.3%)	126,431 (15.1%)	0.13	0.006	0.13
Hypertension	3269 (25.8%)	8152 (21.5%)	179,777 (21.5%)	0.1	0.001	0.1
COPD	342 (2.7%)	691 (1.8%)	14,406 (1.7%)	0.06	0.008	0.06
Chronic Kidney Disease	754 (6.0%)	1760 (4.6%)	37,922 (4.5%)	0.06	0.006	0.05
Cardiovascular Disease	952 (7.5%)	2146 (5.7%)	48,439 (5.8%)	0.07	0.005	0.07
Asthma	2019 (16.0%)	4302 (11.3%)	93,019 (11.1%)	0.14	0.008	0.14

**Table 3 diagnostics-15-03183-t003:** Cox proportional hazard model on the inverse probability weighted data for the specified periods and the entire period. HR, hazard ratio. CI, confidence interval.

	**Stroke**	**Stroke**	**TCI**	**TCI**
	**Hospitalized COVID+ vs. COVID−**	**Non-Hospitalized COVID+ vs. COVID−**	**Hospitalized COVID+ vs. COVID−**	**Non-Hospitalized COVID+ vs. COVID−**
Up to 12 months	1.58 [1.28–1.97]	1.14 [0.95–1.37]	1.09 [0.65–1.82]	2.36 [1.88–2.97]
Up to 24 months	1.45 [1.20–1.75]	1.19 [1.03–1.37]	1.02 [0.71–1.47]	2.31 [1.93–2.76]
Up to 36 months	1.39 [1.18–1.65]	1.16 [1.02–1.33]	1.05 [0.78–1.41]	2.11 [1.78–2.49]
Entire period (Up to 47.7 months)	1.32 [1.12–1.55]	1.21 [1.05–1.39]	1.00 [0.75–1.33]	2.15 [1.81–2.56]

## Data Availability

The data presented in this study are available on request from the corresponding author due to privacy and ethical restrictions.

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
