# Peer review of "Risks of Stroke and Transient Cerebral Ischemia up to 4 Years Post-SARS-CoV-2 Infection in Large Diverse Urban Population in the Bronx"

_diagnostics, 2025, doi:10.3390/diagnostics15243183_

Round 1

Reviewer 1 Report

Comments and Suggestions for Authors
  1. In "Introduction", please provide more information on the association between acute COVID-19, mechanisms of vascular injury, and  stroke or transient cerebral ischemia (TCI)
  2. In "Introduction", consider adding a brief rationale why IPW was chosen to balance covariates between groups.
  3. In "Materials and Methods", please provide clearer information on whether electronic health records (EHR) contain TOAST classification criteria to describe IS subtype.
  4. EHR include patients' data from February 1, 2020 to January 12, 2024. During this period, SARS-CoV-2 virus mutated several times. Was the information on distinct variants of SARS-CoV-2 virus available in EHR?
  5. In "Results", if available, consider adding information on the association between COVID-19 severity (SOFA score, lung CT scan grades 1-4, mechanical ventilation, oxygen supply, etc.) and stroke outcome.
  6. In "Results", data in suppl. file, forest plot B deserves critical analysis. Please, discuss possible reasons for these results, including bias.
  7. In "Discussion", consider discussing elevated D-dimer and ferritin levels as possible triggers of delayed vascular injury and ischemic stroke/TCI.
  8. Discussion would benefit from analyzing the association between SARS-CoV-2 virus variant and risk and severity of ischemic stroke.
  9. In "Discussion", consider analyzing the association between long COVID and ischemic stroke/TCI.
  10. Better avoid using the term "permanent infarction" (line 266). To describe the infarcted area in ischemic stroke, preferred expressions are "permanent ischemia", "brain tissue necrosis", "cerebral infarction".

Author Response

  1. In "Introduction", please provide more information on the association between acute COVID-19, mechanisms of vascular injury, and stroke or transient cerebral ischemia (TCI)

Thank you for the comment. It is addressed now. Following is added

“During acute COVID-19, direct viral invasion of endothelial cells, cytokine-mediated inflammation, and disruption of the renin–angiotensin–aldosterone system can compromise vascular integrity and cerebral perfusion. These changes occur alongside marked coagulopathy—elevated D-dimer, endothelial activation, and microvascular thrombosis—all of which create favorable conditions for ischemic stroke or transient cerebral ischemia. Such mechanisms underscore why SARS-CoV-2 may act as a precipitating factor for acute and possibly persistent cerebrovascular injury.”

  1. In "Introduction", consider adding a brief rationale why IPW was chosen to balance covariates between groups.

Thank you for the comment. It is addressed now. Following is added,

“We selected inverse probability weighting because it provides efficient covariate balance across the three study groups: hospitalized COVID-19+, non-hospitalized COVID-19+, and COVID-19– controls, without reducing sample size, in contrast to propensity matching.”

  1. In "Materials and Methods", please provide clearer information on whether electronic health records (EHR) contain TOAST classification criteria to describe IS subtype.

Thank you for the comment. TOAST classification of ischemic stroke subtype is not available in our EHR (OMOP) dataset.

  1. EHR include patients' data from February 1, 2020 to January 12, 2024. During this period, SARS-CoV-2 virus mutated several times. Was the information on distinct variants of SARS-CoV-2 virus available in EHR?

Thank you for the comment. SARS-CoV-2 strains were not tested on individual patients. The prevalence of variants was based on population level, and there were significant overlaps of the predominant variants at any given time, and thus defining predominant variants that patients were affected over a period is challenging, somewhat arbitrary and inaccurate. We thus did not analyze outcomes with respect to variants of the SARS-CoV-2 virus. Distinct variants of the SARS-CoV-2 virus are not available in the EHR.

  1. In "Results", if available, consider adding information on the association between COVID-19 severity (SOFA score, lung CT scan grades 1-4, mechanical ventilation, oxygen supply, etc.) and stroke outcome.

Thank you for the comment. SOFA score and lung CT are not documented in the EMR. We agree that mechanical ventilation status is important and patients who had IMV are likely to have a higher incidence of PF. During the early pandemic, mechanical ventilation status, how patients were treated and oxygen treatment were not accurately or consistently coded. We thus did not analyze outcomes with respect to these variables. We only use COVID-19 hospitalization status as indicator of COVID-19 severity.

  1. In "Results", data in suppl. file, forest plot B deserves critical analysis. Please, discuss possible reasons for these results, including bias.

Thank you for your comments. It is addressed now. We have added the following to the result.

“Across most subgroups, non-hospitalized COVID-19+ patients showed an increased risk of TCI compared with controls, and increased stroke risk was observed among those younger than 60, female, Hispanic, insured through Medicare, or with hypertension, asthma, or chronic kidney disease. In contrast, hospitalized COVID-19+ patients showed higher stroke risk only in select subgroups, including individuals older than 60, males and females, Black and Hispanic patients, those covered by Medicaid, and those with hypertension or asthma. Nearly all subgroups demonstrated no association between hospitalized COVID-19 positivity and TCI. ”

These trends are consistent with the main analysis. In the Discussion section, we have added text explaining why non-hospitalized COVID-19–positive patients show a higher risk of TCI, whereas hospitalized COVID-19–positive patients do not demonstrate this association.

“The higher hazard ratio for TCI in non-hospitalized COVID-19 survivors, contrasted with the null association in the hospitalized group, appears counterintuitive. Several factors may explain this pattern. First, hospitalized patients with severe COVID-19 often receive early anti-inflammatory and antithrombotic therapies, which may mitigate subsequent vascular injury and reduce the likelihood of post-acute cerebrovascular events. Second, severe COVID-19 is associated with high short-term mortality; individuals at greatest risk for cerebrovascular complications may not have survived long enough to develop TCI, creating a competing-risk bias that attenuates associations in the hospitalized group. Third, diagnostic and coding patterns may differ between groups: hospitalized patients typically undergo comprehensive inpatient neurological evaluation and may have residual symptoms managed under existing diagnoses, whereas non-hospitalized patients who later present with transient focal deficits are more likely to receive a new TCI code. Finally, the number of TCI events was modest, and given multiple subgroup analyses, some of the observed heterogeneity may reflect chance. Together, these explanations provide a plausible context for the observed differences while acknowledging the limitations inherent to EHR-based analyses.”

  1. In "Discussion", consider discussing elevated D-dimer and ferritin levels as possible triggers of delayed vascular injury and ischemic stroke/TCI.

Thank you for your comments. It is addressed now. We have added the following to the discussion,

“Elevated D-dimer and ferritin levels observed during acute COVID-19 may contribute to ongoing vascular injury and prothrombotic activity, providing a plausible mechanism for delayed ischemic stroke or TCI (49, 50).”

  1. Discussion would benefit from analyzing the association between the SARS-CoV-2 virus variant and risk and severity of ischemic stroke.

Thank you for your comment. It is addressed now. We have added following to the discussion,

“The risk of ischemic stroke may vary across SARS-CoV-2 variants, reflecting differences in viral pathogenicity and the degree of associated inflammation and coagulopathy. Early variants such as Alpha and Delta were linked to higher rates of acute thrombotic complications, including stroke, likely due to more pronounced endothelial injury and hypercoagulability. In contrast, Omicron has generally shown lower acute neurological and thrombotic involvement, despite its higher transmissibility. These variant-specific differences underscore the evolving cerebrovascular impact of COVID-19 over the course of the pandemic (33-35). However, it is important to note that the effects of variants to outcomes were likely confounded by available vaccines, available treatments, and other factors and are difficult to segregate.”

  1. In "Discussion", consider analyzing the association between long COVID and ischemic stroke/TCI.

Thank you for your comments. Long covid includes a wide range of medical conditions. We did not analyze the association between long COVID and ischemic stroke/TCI.

  1. Better avoid using the term "permanent infarction" (line 266). To describe the infarcted area in ischemic stroke, preferred expressions are "permanent ischemia", "brain tissue necrosis", "cerebral infarction".

Thank you for the comment. It is addressed now.

Reviewer 2 Report

Comments and Suggestions for Authors

Risks of stroke and transient cerebral ischemia up to 4 years post SAR-CoV-2 infection in large diverse urban population in the Bronx

This is an interesting study investigates the risk of having stroke 4 years after COVID infection. It would have been informative to stratify stroke incidence according to time from infection. This would clarify if strokes/TCI were temporally related to acute infection or not.
Another important point to clarify, did these patients enter hospital due to some disease other than COVID, or were they healthy and just screened for COVID?

Materials and Methods:
-    71-74: did these patients enter hospital due to some disease or were they healthy and just screened for COVID?
Results:
-    127: need to identify your 3 groups in this text so that reader can understand what the 3 numbers in the comparison refer to.
-    Table 1: if you added the p values for the intergroup difference this would be more informative about the significance of this difference.
-    150: does this mean that there was no significant diff between groups?
-    153: according to what was stated in study cohort, patients with a history of cerebrovascular events are supposed to be excluded from the start.
Discussion:
-    212: grammatic revision
-    214: I suppose this phrase needs language edit
-    214-229: This paragraph is not consistent, you started by stating that there was no increased risk of TCI in hospitalized COVID cases compared to controls, then the rest of the paragraph is rationalizing an increased TCI risk and that it is more sensitive than stroke?
If you want to find a rationale for non hospitalized+ having higher risk than hospitalized +, then it might be that those hospitalized possibly received anti-inflammatory medications in the early stage so they might be less prone to vascular sequel of inflammation.

Author Response

Risks of stroke and transient cerebral ischemia up to 4 years post SAR-CoV-2 infection in large diverse urban population in the Bronx

This is an interesting study investigates the risk of having stroke 4 years after COVID infection.

  1. It would have been informative to stratify stroke incidence according to time from infection. This would clarify if strokes/TCI were temporally related to acute infection or not.

Thank you for the comments. We have added the time stratified hazard ratio to Table 3B.

  1. Another important point to clarify, did these patients enter hospital due to some disease other than COVID, or were they healthy and just screened for COVID?

Thank you for your comments. The following is added.

“Data came from the electronic health records (EHR) of the Montefiore Health System, which consists of multiple hospitals and outpatient clinics in the Bronx and its environs. Patents came to our healthy system for any medical reasons, including regular check-ups and screening for COVID-19.”

  1. Materials and Methods:
     71-74: did these patients enter hospital due to some disease or were they healthy and just screened for COVID?

Thank you for your comments. The following is added.

“Data came from the electronic health records (EHR) of the Montefiore Health System, which consists of multiple hospitals and outpatient clinics in the Bronx and its environs. Patents came to our healthy system for any medical reasons, including regular check-ups and screening for COVID-19.”

  1. Results: 127: need to identify your 3 groups in this text so that reader can understand what the 3 numbers in the comparison refer to.

Thank you for the comment. It is addressed now and following is added to the results.

“Table 1 shows the baseline characteristics of the three study groups: hospitalized COVID-19+, non-hospitalized COVID-19+, and COVID-19– patients.”

  1. Table 1: if you added the p values for the intergroup difference this would be more informative about the significance of this difference.

Thank you for your comment. We report standardized mean differences (SMDs) to assess intergroup balance, which is the preferred approach when using inverse probability weighting, as p-values are not as appropriate for weighted pseudo-populations.

  1. 150: does this mean that there was no significant diff between groups?

Thank you for your comments. Table 2 presents the weighted pseudopopulation, in which most variables achieved a standardized mean difference below 0.1, indicating that covariates were well balanced and no meaningful differences remained between groups after weighting.

  1. 153: according to what was stated in study cohort, patients with a history of cerebrovascular events are supposed to be excluded from the start.

Thank you for your comment. As described in the Methods, patients with a prior history of cerebrovascular events were excluded before cohort construction, and Table 1 reflects the final cohort after this exclusion.

  1. Discussion:
    -    212: grammatic revision

Thank you. Corrected
-    214: I suppose this phrase needs language edit

Thank you. Corrected
-    214-229: This paragraph is not consistent, you started by stating that there was no increased risk of TCI in hospitalized COVID cases compared to controls, then the rest of the paragraph is rationalizing an increased TCI risk and that it is more sensitive than stroke?
If you want to find a rationale for non hospitalized+ having higher risk than hospitalized +, then it might be that those hospitalized possibly received anti-inflammatory medications in the early stage so they might be less prone to vascular sequel of inflammation.

Thank you for your thoughtful comments. We agree with you and we attempted to explain. Thank you for your suggestions on the possible explanation. The following is added.

The higher hazard ratio for TCI in non-hospitalized COVID-19 survivors, contrasted with the null association in the hospitalized group, appears counterintuitive. Several factors may explain this pattern. First, hospitalized patients with severe COVID-19 often receive early anti-inflammatory and antithrombotic therapies, which may mitigate subsequent vascular injury and reduce the likelihood of post-acute cerebrovascular events. Second, severe COVID-19 is associated with high short-term mortality; individuals at greatest risk for cerebrovascular complications may not have survived long enough to develop TCI, creating a competing-risk bias that attenuates associations in the hospitalized group. Third, diagnostic and coding patterns may differ between groups: hospitalized patients typically undergo comprehensive inpatient neurological evaluation and may have residual symptoms managed under existing diagnoses, whereas non-hospitalized patients who later present with transient focal deficits are more likely to receive a new TCI code. Finally, the number of TCI events was modest, and given multiple subgroup analyses, some of the observed heterogeneity may reflect chance. Together, these explanations provide a plausible context for the observed differences while acknowledging the limitations inherent to EHR-based analyses.

Reviewer 3 Report

Comments and Suggestions for Authors

This retrospective cohort study by Changela et al. investigates the long-term risks of ischemic stroke and transient cerebral ischemia (TCI) in patients with SARS-CoV-2 infection compared to controls within the Montefiore Health System in the Bronx (2020–2024). The study utilizes a large dataset (over 52,000 COVID+ and 837,000 COVID- patients) and employs Inverse Probability Weighting (IPW) to adjust for significant demographic and socioeconomic confounders. Please consider my suggestion listed below for improvement:

  1. There is a significant mismatch between the Hazard Ratios (HR) reported in the abstract and those reported in table 3. Please clarify which one is correct.
  2. The finding that TCI risk is doubled (HR > 2.0) in non-hospitalized (mild) patients but non-existent (HR 1.00) in hospitalized (severe) patients contradicts the biological plausibility that severe infection leads to greater vascular injury. Please elaborate if this phenomenon was because of surveillance bias or ascertainment bias.
  3. Is there any possibility of the presence of undefined COVID (+) subjects? Please discuss it.
  4. The study excludes patients lost to follow-up within 14 days of the index date. While this is standard for long-term studies, it excludes patients who died of stroke or had massive cerebrovascular events in the acute phase (days 0–14). Therefore, the study does not measure the total risk of stroke from COVID-19, but rather the risk in survivors of the acute phase. This distinct must be stated. 
  5. Please re-format the header and row of your table according to author's instruction.
  6. The methods mention using the Health Leads Toolkit. Please clarify what percentage of the population actually completed this voluntary screening.
  7. The title contains a significant typographical error: "SAR-CoV-2". This must be corrected to "SARS-CoV-2
  8. Figure 2 appears to have low resolution, and the risk tables below the x-axis are difficult to read. Please provide high-resolution vector graphics.
  9. The reference list has inconsistent formatting. Please check.

Author Response

This retrospective cohort study by Changela et al. investigates the long-term risks of ischemic stroke and transient cerebral ischemia (TCI) in patients with SARS-CoV-2 infection compared to controls within the Montefiore Health System in the Bronx (2020–2024). The study utilizes a large dataset (over 52,000 COVID+ and 837,000 COVID- patients) and employs Inverse Probability Weighting (IPW) to adjust for significant demographic and socioeconomic confounders. Please consider my suggestion listed below for improvement:

  1. There is a significant mismatch between the Hazard Ratios (HR) reported in the abstract and those reported in table 3. Please clarify which one is correct.

Thank you for your comments. It is addressed now.

  1. The finding that TCI risk is doubled (HR > 2.0) in non-hospitalized (mild) patients but non-existent (HR 1.00) in hospitalized (severe) patients contradicts the biological plausibility that severe infection leads to greater vascular injury. Please elaborate if this phenomenon was because of surveillance bias or ascertainment bias.

Thank you for your comment. This is added

“The higher hazard ratio for TCI in non-hospitalized COVID-19 survivors, contrasted with the null association in the hospitalized group, appears counterintuitive. Several factors may explain this pattern. First, hospitalized patients with severe COVID-19 often receive early anti-inflammatory and antithrombotic therapies, which may mitigate subsequent vascular injury and reduce the likelihood of post-acute cerebrovascular events. Second, severe COVID-19 is associated with high short-term mortality; individuals at greatest risk for cerebrovascular complications may not have survived long enough to develop TCI, creating a competing-risk bias that attenuates associations in the hospitalized group. Third, diagnostic and coding patterns may differ between groups: hospitalized patients typically undergo comprehensive inpatient neurological evaluation and may have residual symptoms managed under existing diagnoses, whereas non-hospitalized patients who later present with transient focal deficits are more likely to receive a new TCI code. Finally, the number of TCI events was modest, and given multiple subgroup analyses, some of the observed heterogeneity may reflect chance. Together, these explanations provide a plausible context for the observed differences while acknowledging the limitations inherent to EHR-based analyses.”

  1. Is there any possibility of the presence of undefined COVID (+) subjects? Please discuss it.

Thank you for your comments. It is addressed now. The following is added to the limitation,

“We restricted inclusion to PCR-confirmed COVID-19 cases, as home test results were inconsistently documented, and some patients may have tested positive outside our system. Consequently, misclassification of COVID-19 status may have occurred, with undetected infections being included in the COVID-negative group. This would likely bias our effect estimates toward the null.”

  1. The study excludes patients lost to follow-up within 14 days of the index date. While this is standard for long-term studies, it excludes patients who died of stroke or had massive cerebrovascular events in the acute phase (days 0–14). Therefore, the study does not measure the total risk of stroke from COVID-19, but rather the risk in survivors of the acute phase. This distinct must be stated. 

Thank you for your comments. It is addressed now. We have stated this in the method.

  1. Please re-format the header and row of your table according to author's instruction.

Thank you for your comments.

  1. The methods mention using the Health Leads Toolkit. Please clarify what percentage of the population actually completed this voluntary screening.

Thank you for your comments. Apologies for the confusion. The Health Leads Toolkit is irrelevant as we are not using the social determinant of health variable.

  1. The title contains a significant typographical error: "SAR-CoV-2". This must be corrected to "SARS-CoV-2

Thank you for your comments. It is addressed now.

  1. Figure 2 appears to have low resolution, and the risk tables below the x-axis are difficult to read. Please provide high-resolution vector graphics.

Thank you for your comments. It is addressed now.

  1. The reference list has inconsistent formatting. Please check.

Thank you for your comments. It is addressed now.

Round 2

Reviewer 1 Report

Comments and Suggestions for Authors

No additional comments or suggestions

Author Response

Thank you so much.